# Antioxidant and Antidiabetic Activity of *Cornus mas* L. and *Crataegus monogyna* Fruit Extracts

**DOI:** 10.3390/molecules29153595

**Published:** 2024-07-30

**Authors:** Gabriela Paun, Elena Neagu, Camelia Albu, Andreia Alecu, Ana-Maria Seciu-Grama, Gabriel Lucian Radu

**Affiliations:** National Institute for Research-Development of Biological Sciences, Centre of Bioanalysis, 296 Spl. Independentei, P.O. Box 17-16, 060031 Bucharest, Romania; elena.neagu@incdsb.ro (E.N.); camelia.albu@incdsb.ro (C.A.); andreia.alecu@incdsb.ro (A.A.); anamaria.seciu@incdsb.ro (A.-M.S.-G.)

**Keywords:** green extraction, antioxidant, antidiabetic, *Cornus mas* L., *Crategus monogyna*

## Abstract

The present study evaluated three green extraction methods, accelerated solvent extraction (ASE), ultrasound-assisted extraction (UAE), and laser irradiation extraction (LE), for the polyphenolic compounds and vitamin C extraction of *Cornus mas* L. and *Crataegus monogyna* fruit extracts. The polyphenols and vitamin C of extracts were quantified using HPLC-DAD, and the total phenolic content, flavonoid content, antioxidant activity (DPPH and reducing power), and antidiabetic activity were also studied. The antidiabetic activity was examined by the inhibition of α-amylase and α-glucosidase, and in vitro on a beta TC cell line (β-TC-6). The results showed significant differentiation in the extraction yield between the methods used, with the ASE and LE presenting the highest values. The *C. mas* fruit extract obtained by ASE exhibited the best antioxidant activity, reaching an IC_50_ value of 31.82 ± 0.10 µg/mL in the DPPH assay and 33.95 ± 0.20 µg/mL in the reducing power assay. The *C. mas* fruit extracts obtained by ASE and LE also have the highest inhibitory activity on enzymes associated with metabolic disorders: α-amylase (IC_50_ = 0.44 ± 0.02 µg/mL for the extract obtained by ASE, and 0.11 ± 0.01 µg/mL for the extract obtained by LE at combined wavelengths of 1270 + 1550 nm) and *α*-glucosidase (IC_50_ of 77.1 ± 3.1 µg/mL for the extract obtained by ASE, and 98.2 ± 4.7 µg/mL for the extract obtained by LE at combined wavelengths of 1270 + 1550 nm). The evaluation of in vitro antidiabetic activity demonstrated that the treatment with *C. mas* and *C. monogyna* fruit extracts obtained using ASE stimulated the insulin secretion of β-TC-6 cells, both under normal conditions and hyperglycemic conditions, as well. All results suggest that *C. mas* and *C. monogyna* fruit extracts are good sources of bioactive molecules with antioxidant and antidiabetic activity.

## 1. Introduction

In recent decades, there has been increased interest in the traditional use of medicinal plants in the prevention of many diseases and associated complications. The most important criteria justifying these trends are safety, effectiveness, economic feasibility, and accessibility [1]. The secondary metabolites present in medicinal plants, fruits, and vegetables, especially from the class of polyphenols, contribute significantly to the therapeutic effectiveness of medicinal plants [2]. Numerous studies demonstrated their antioxidant, anti-inflammatory, antimicrobial, and anticancer activities and their ability to modulate critical cellular enzyme functions [3,4,5].

Diabetes is a major health due to its increasing prevalence, also being a significant factor in both mortality and morbidity worldwide. One way to treat diabetes is by inhibiting the digestive enzymes α-amylase and α-glucosidase involved in the degradation of carbohydrates [6]. Classic antidiabetic drugs such as metformin and acarbose have side-effects such as nausea, weakness, and bloating [7]; others such as sulphonylureas increase the risk of mortality from cardiovascular diseases [8]. That is why natural alternatives are sought, such as biologically active compounds from medicinal plants, with antidiabetic effects and less negative impacts on health [9]. These bioactive compounds have scarce side effects compared to synthetic drugs, and recent research demonstrated the importance of the synergistic effect of bioactive compounds in a natural intermixture.

Oxidative stress, resulting from an inequality between reactive oxygen species production and antioxidant defense, contributes to the pathogenesis of diabetes and its complications [10]. Antioxidants from natural sources have shown promise in reducing oxidative stress and antidiabetic potential. Numerous researchers have pointed out that bioactive compounds with antioxidant activity can inhibit the enzymes involved in carbohydrate hydrolysis and could be beneficial in managing diabetes mellitus [11].

Research has demonstrated a good choice of extraction method, and solvents influence the content of target bioactive compounds in the extract, leading to increased product effectiveness. There is a high interest in green and sustainable extraction technologies due to their advantages, such as a shorter extraction time, higher selectivity, lower organic solvent, and the preservation of valuable bioactive compounds [12,13,14,15]. Among these technologies, ultrasound-assisted (UAE) and accelerated solvent extraction (ASE) have gained more attention in recent years [13,14,15,16,17]. Recently, laser irradiation extraction (LE) has aroused researchers’ interest, demonstrating the ability of selective extraction for certain polyphenolic compounds [18,19]. UAE and ASE techniques are green and economically viable approaches for extracting natural products. Recent studies demonstrated that UAE and ASE have a high efficiency in extracting bioactive compounds, leading to high yields of biologically active compounds with low or minimal damage to their biological properties, as well as the production of little waste in the extraction process. Novel and efficient laser extraction technology is based on the destruction of vegetal tissues caused by the intensive vapor which is formed via the laser energy absorbed by water contained in the biological matrix [20]. Compared with ASE, a major advantage of laser-assisted extraction lies in the shorter processing time for processing the same extract volume.

*Cornus mas* L. and *Crataegus monogyna* are two species that are very widespread in Romania, but they have been exploited very little. In addition, we studied them together based both on their high content of vitamin C and anthocyanin compounds, as well as previous studies that highlighted their antidiabetic activity in vivo. *Cornus mas* L. (Cornelian cherry) belongs to the Cornaceae family and grows in Europe and Southwest Asia [21,22]. The fruits have been used for decades in many European and Asian countries both in traditional cuisine and in folk medicine. Recently, studies showed that *C. mas* fruits have anti-inflammatory, antibacterial, antioxidant, antidiabetic, antiproliferative, antiparasitic, antihyperlipidemic, hepatoprotective, and neuroprotective effects [22,23,24,25,26,27,28]. The fruits contain high levels of anthocyanins, iridoids, polyphenols, vitamin C, minerals, tannins, sugars, pectin, and organic acids, while the leaves contain significant amounts of phenolic acids [29].

*Crataegus monogyna* (hawthorn) is a valuable medicinal plant that is part of the Rosaceae family. Making part of the European flora, hawthorn is widely spread [30]. The plant is used in traditional medicine to treat digestive, microbial, and cardiac problems, due to its antispasmodic, cardiotonic, antiatherosclerotic, and hypotensive properties and diabetes disorders [31]. *C. monogyna* is a source of bioactive compounds, mainly polyphenols, like epicatechin, procyanidins, isoquercitrin, vitexin, and chlorogenic acid, but also triterpenic acids, such as ursolic acid and oleanolic acid, carboxylic acids, and minerals [9,32,33]. Among these compounds, epicatechin, the most abundant component in hawthorn fruits, has been found to elevate insulin sensitivity and lower insulin resistance [34].

Despite some data on the in vivo antidiabetic effects of *C. mas* and *C. monogyna* fruits related to the reduction in blood glucose levels and the increase in insulin levels in diabetic rats, there are very few studies related to the inhibition of enzymes related to metabolic diseases [30,35,36].

Accordingly, the main goal of the current research was to evaluate three green extraction methods (accelerated solvent extraction, ultrasound-assisted extraction, and laser irradiation extraction), determine the content of selected phytochemicals (polyphenolic compounds and vitamin C) in *C. mas* and *C. monogyna* fruit extracts, and examine the antioxidant and antidiabetic activities by α-amylase and α-glucosidase inhibition of these fruit extracts. An in vitro antidiabetic study for the most effective extracts in terms of inhibiting carbohydrate-digesting enzymes was carried out, but also on a murine insulinoma cell line (β-TC-6).

## 2. Results and Discussion

### 2.1. Influence of Extraction Technique on Bioactive Compounds Content

First, we obtained the extracts of *C. mas* and *C. monogyna* fruits using three green extraction techniques, ASE, UAE, and LE, and then compared the content of polyphenolic compounds to obtain the maximum yield of bioactive compounds (Table 1).

ASE was shown as the most effective green extraction method to extract the targeted compounds from *C. mas* (103.94 ± 2.6 mg/g chlorogenic acid equivalent for total polyphenols and 2.27 ± 0.02 mg/g rutin equivalent for flavonoid content) and from *C. monogyna* (84.89 ± 0.6 mg/g chlorogenic acid equivalent for total polyphenols and 3.02 ± 0.03 mg/g rutin equivalent for flavonoid content) compared to the other techniques. As noticeable, UAE showed the lowest yield %, while for ASE, the highest yield values were obtained. At the same time, LE at combined wavelengths (1550 nm and 1270 nm) demonstrated a higher efficiency compared to UAE for extracting the target compounds. Another study proved significant increases in total phenolics from extracts by laser irradiation at 785 nm (+15.50), while at lengths lower than 550 nm, the extraction was inefficient [18]. Although it is a very new method, its effectiveness was shown in a previous study by the same team of researchers [19].

ASE, involving using solvents at high pressure and temperature, accelerates the kinetics of the extraction process, leading to fast and safe extractions. The high performance of ASE depends on several parameters such as solvent, temperature, the number of extraction cycles, and the static time. Some conditions such as heat and extraction time can degrade polyphenolic compounds and bioactivity. This is why the extraction was carried out in three cycles at 80 °C. Another research study demonstrated that the optimal ASE parameters for polyphenol extraction were 110 °C, 3 extraction cycles, and 10 min of static time [37].

In UAE, the solvent extracts components from plants using sound energy as the primary driving mechanism representing an efficient, environmentally friendly method [38]. Several parameters, such as frequency, temperature, time, type of solvent, and solid–liquid ratio, also influence the extraction. For a better comparison of the effectiveness of the three methods approached, the following parameters were kept constant: the same solvent (50% (*v*/*v*) hydroalcoholic solution), extraction time of 30 min, extraction temperature of 80 °C, and solid–liquid ratio of 1:0.9. The use of a hydroethanolic solution was chosen because there are differences in the solubility of the extracted polyphenolic compounds, where some have a higher solubility in water, while others are extracted with a higher yield in ethanol.

The main mechanism of LE can be a photochemical process, thermal process, or a combination of both. Thermal effects result in the conversion of light into heat, followed by heat transfer into the irradiated tissue leading to melting or liquid vaporization in vegetal plants, thus leading to the extraction of compounds from the vegetal matrix. The photochemical effects depend on the wavelength used (nm), and the chemical bonds of molecules present in the substrate are broken by absorbing the high photon energy, leading to a series of chemical reactions, like reactive (oxygen) species formation [18,20]. In the specific case of vegetal extraction, laser irradiation is used to intensify biochemical reactions and biomass accumulation in the medium, intensifying heat and mass-exchange processes in the medium and increasing the amount of specific compounds in the final extract. LE can perform selective processing due to the absorption difference in laser energy for different compounds.

HPLC-DAD quantified phenolic compounds, anthocyanins, and vitamin C for each extract. The extraction method’s influence on each analyte’s content was determined (Table 2 and Table 3), and the performance characteristics of the HPLC-DAD method are presented in Table 4. The chromatograms for *C. mas* and *C. monogyna* extracts obtained by ASE are presented in Figure 1 and Figure 2.

The results proved that laser irradiation is an effective extraction technique for target compounds, and ASE ensures efficient extraction of the desired compounds. LE is a very new extraction method, which demonstrated the efficiency in the extraction of phenolic acids, flavonoids, and isoflavonoids from plants at 552 nm, 660 nm, 785 nm, and at a combined 1270 and 1550 nm [18,19]. Laser extraction is very poorly studied, but the present study demonstrates effectiveness in the extraction, especially of anthocyanidins and flavonoids.

Several research studies have indicated that *C. mas* fruit is a rich source of vitamin C and anthocyanins. Martinović and Cavoski [39] reported 48–108 mg/100 g vitamin C and 158–591 mg GAE/100 g total polyphenol. In research performed by Moldovan et al. [40], the main polyphenols were ellagic acid (187.91 mg/100 g), followed by chlorogenic (32.76 mg/100 g) and caffeic acid (27.12 mg/100 g). In our study, the most abundant phenolic acids were caffeic acid (701.63–1020.56 µg/g), followed by chlorogenic acid (362.09–504.72 µg/g). At the same time, rutin was the main flavonoid compound and cyanidin-3-glucopyranoside was the main anthocyanin compound. Among anthocyanins, in our extracts, cyanidin-3-glucopyranoside was found in the largest amount, while Moldovan et al. [40] found pelargonidin-3-O-glucoside as the anthocyanin with the highest concentration.

However, the results obtained in this research differ from those obtained by the studies mentioned above, which are most likely related to the extraction method, region of the cultivar, abiotic factors, stages of maturity, and other influences.

The HPLC–DAD analysis of *C. monogyna* fruit extracts showed a significant catechin and epicatechin content. Some recent studies have highlighted the presence of isoquercitrin, vitexin, quercetin, rutin, quercetin-3-glucoside, caffeic acid, (−)- epicatechin, and vitamin C in berries of all hawthorn species, inclusive in *C. monogyna* species [30,41]. Our data confirmed those of previously published studies reporting significant amounts of rutin, (+)-catechin, epicatechin, and vitamin C. To our knowledge, delphinidin-3-glucopyranoside and delphinidin were not reported previously in *C. monogyna* fruit extracts.

The results indicate that the fruits of *C. mas* and *C. monogyna* are a rich source of total polyphenols, anthocyanins, and vitamin C, exhibiting various biological effects. This is the first comparative study of obtaining extracts from C. mas and C. monogyna through the ASE and LE methods and obtaining extracts enriched in polyphenols and vitamin C.

### 2.2. Total Antioxidant Activity

Due to the complex activity of the phytoconstituents, the antioxidant activity of plant extracts cannot be assessed by a single method, and it is advisable to use several methods, as each technique provides complementary information. In this regard, the antioxidant activity of the *C. mas* and *C. monogyna* extracts was determined by the DPPH radical scavenging activity and reducing power assays. Our results of the values of the IC_50_ parameter are presented in Table 5. The inhibition curves for DPPH inhibition are presented in Appendix A.

As shown, the *C. mas* fruit extract obtained by ASE exhibited the best antioxidant activity, reaching an IC_50_ value of 31.82 ± 0.1 µg/mL in the DPPH assay and 33.95 ± 0.2 µg/mL in the reducing power assay. The IC_50_ values related to the DPPH and reducing power assays for all *C. mas* fruit extracts lower than vitamin C showed high antioxidant activity. At the same time, all extracts obtained by laser irradiation presented a higher antioxidant activity than those obtained by UAE. The main antioxidant components of the studied fruit extracts are phenolic compounds, anthocyanins, and vitamin C, and their total content is proportional to the antioxidant activity.

Blagojević et al. [42] studied the antioxidant activity of *C. mas* hydroalcoholic extracts and the results showed that the extracts presented a high DPPH inhibition (IC_50_ in the range of 1.76–3.59 mg/mL) and ferric-reducing power (7.83–23.95 mg/g). In our study, the extracts had a significantly higher antioxidant capacity (IC_50_ = 31.82 ± 0.1 ÷ 54.70 ± 0.4 µg/mL in the DPPH assay), with the explanation being the efficiency of the applied extraction methods [42].

The inhibition of free radicals may be the result of the action of various phenolic acids detected in the studied extracts, and in higher content in *C. mas* extracts such as gallic acid, chlorogenic acid, caffeic acid, and coumaric acid, but also other polyphenolic compounds not quantified in the studied extracts. The literature data indicate that polyphenols can inhibit cell destruction by free radicals and suppress enzymes involved in intracellular ROS overproduction [43,44].

The antioxidant effect of the studied extracts is also related to vitamin C, the well-known antioxidant compound that manifests its antioxidant properties in living cells by directly scavenging free radicals, activating intracellular antioxidant systems, or supporting the action of other exogenous antioxidants [44].

### 2.3. Antidiabetic Activity

α-amylase and α-glucosidase inhibition

The inhibitory effects of *C. mas* and *C. monogyna* extract on α-amylase and α-glucosidase took place in a dose-dependent manner. The IC_50_ values of all extracts against these enzymes are presented in Table 6. The inhibition curves for α-amylase and α-glucosidase inhibition tests are presented in Appendix A.

The inhibition of α-amylase and α-glucosidase slows down the hydrolysis of α-1,4-glucan polysaccharides such as starch, which helps reduce the levels of glucose in the blood. The α-amylase and α-glucosidase inhibitors could be used for new drug developments in treating type 2 diabetes. The *C. mas* fruit extracts obtained by ASE and laser irradiation (LE) have a higher inhibitory activity on both enzymes—α-amylase (IC_50_ = 0.44 ± 0.02 µg/mL for the extract obtained by ASE, and 0.11 ± 0.01 µg/mL for the extract obtained by LE at combined wavelengths of 1270 + 1550 nm) and α-glucosidase (IC_50_ of 77.1 ± 3.1 µg/mL for the extract obtained by ASE, and 98.2 ± 4.7 µg/mL for the extract obtained by LE at combined wavelengths of 1270 + 1550 nm)—being lower than on acarbose, used as the standard compound. This is in agreement with previous studies which showed a high inhibitory effect of *C. mas* fruit extract [45].

The *C. monogyna* extracts obtained by ASE and LE also significantly inhibited α-amylase with IC_50_ = 0.53 ± 0.01 µg/mL for the extract obtained by ASE, and 1.26 ± 0.1 µg/mL for the extract obtained by LE at combined wavelengths of 1270 + 1550 nm, compared to acarbose with IC_50_ values of 8.12 ± 0.6 µg/mL. The data obtained showed a good inhibition of *C. monogyna* ASE and LE extracts on *α*-glucosidase, also.

However, the amount of this phenolic compound present in the ASE may be partially responsible for the inhibition of α-amylase and α-glucosidase. Hydroxyl (-OH) groups are essential for the inhibitory activity of polyphenolic compounds against α-amylase, as the inhibition is linked to the formation of hydrogen bonds between the -OH groups of phenolics and amino acids residues at the active site of α-amylase [46]. Phenolic acids with a greater number of -OH groups presented a higher inhibition effect against both enzymes [47]. The inhibitory effects of phenolic acids are linked to the number of -OH groups, and chlorogenic and gallic acid have a higher inhibition effect against both enzymes [48]. Other flavonoids and anthocyanins, such as quercitrin, quercetin, myricetin, cyanidin-3-glucopyranoside, and petunidin-3-glucoside present in the extracts from the present study, have been previously reported to have a hypoglycemic effect and stronger inhibitory effect on α-glucosidase [49,50,51].

Insulin secretion assay

In this study, we investigated in vitro the complementary antidiabetic activity of *C. mas* and *C. monogyna* fruit extracts obtained by ASE, demonstrating the highest inhibitory activities on carbohydrate-hydrolyzing enzymes. The extract concentrations were tested (250 and 500 μg/mL) on a murine insulinoma cell line (β-TC-6).

All tested extracts increased insulin concentrations compared to the control, both in normal conditions (5.6 mM) and in hyperglycemic conditions (16.7 mM) (Figure 3).

In normal glycemic conditions, the highest concentration of secreted insulin was obtained after treatment with a concentration of 500 µg/mL, both in the case of *C. mas* extract (121 µU/mL) and in the case of *C. monogyna* extract (134 μU/mL).

In hyperglycemic conditions at 500 μg/mL, both extracts elicited a marked increase in insulin secretion in the β-TC-6 cells, 352 µU/mL for *C. monogyna* extract and 364.86 µU/mL for *C. mas* extract, while for the untreated control, the concentration of secreted insulin was 143 μU/mL.

Correlated with the results obtained for the inhibition of α-amylase and α-glucosidase enzymes, the in vitro studies on the β-TC-6 cell lines demonstrate that both extracts presented a strong stimulatory effect of insulin secretion in the in vitro β-TC-6 pancreatic cell stimulation model, similar to alanine used as a positive control.

These extracts were tested for their hypoglycemic effects especially in diabetic rat models (in vivo). However, this is the first study on the impact of various concentrations of *C. mas* and *C. monogyna* on the insulin section in the β-TC-6 pancreatic cell line.

## 3. Materials and Methods

### 3.1. Materials

All chemicals used in this study were analytical-grade reagents. Chlorogenic acid, p-coumaric acid, caffeic acid, luteolin, (+)-catechin, quercetin 3-glucoside, kaempferol, quercitrin, isorhamnetin, quercetin, rutin, cyanidin, vitamin C (all >96% purity, HPLC), acetic acid (100%), formic acid (>95%), and sodium 1-heptane-sulfonate were bought from Sigma-Aldrich (St. Louis, MO, USA). Ellagic acid (>95%, HPLC), (−)-epicatechin (>96%, HPLC), gallic acid (≥98%, HPLC), and myricetin (>96%, HPLC) were acquired from Fluka (Sigma-Aldrich Chemie GmbH, Schnelldorf, Germany). PhytoLab (PhytoLab GmbH & Co., Vestenbergsgreuth, Germany) provided delphinidin (>90%, HPLC) and petunidin-3-glucoside (>95%, HPLC). Delphinidin-3-glucoside (>97%) and cyanidin-3-glucoside (>97%) were procured from Polyphenols AS (Sandnes, Norway). 2,2-diphenyl-1-picrylhydrazyl (DPPH); potassium ferricyanide (>99%); sodium carbonate (≥99.5%, Na_2_CO_3_); 3,5-dinitrosalicylic acid, 98% (DNS); α-amylase from hog pancreas; α-glucosidase from *Saccharomyces cerevisiae*; acarbose (≥95%, HPLC); and 4-nitrophenyl α-d-glucopyranoside (NPG) were acquired from Sigma–Aldrich (St. Louis, MO, USA). The other used substances, methyl alcohol (Riedel-de Haen, Honeywell Riedel-de Haën, Seelzer, Germany) and ethyl alcohol (analytical reagent, Chemical Company, Iasi, Romania), were of chromatographic or analytical purity. The Mouse Insulin ELISA kit, DMEM culture media, alanine, and glucose were purchased from Sigma-Aldrich (St. Louis, MO, USA). The Penicillin–Streptomycin–Neomycin (PSN) Antibiotic Mixture and fetal bovine serum were acquired from Gibco (Thermo Fisher Scientific, Waltham, MA, USA); CO_2_ (99.98% purity) was purchased from Auto Gri SRL, Romania. Double-distilled water was obtained using the distillation apparatus from Evoqua Water Technologies (Pittsburgh, PA, USA).

Fresh ripe fruits of *C. mas* and *C. monogyna* were harvested from the Breaza area, in Prahova County, Romania (45°11′14″ N, 25°39′44″ E), in September 2023. Fruits were washed with double-distilled water, dried and ground, and subjected to extraction. Drying was performed in an oven at 40 °C (Pol-Eko Aparatura, Wodzisław Śląski, Poland). The dried plant material of each species was finely ground into a powder using a Grindomix GM100 machine (Retsch, Haan, Germany).

### 3.2. Extract Preparation

Three green extraction technologies were applied to investigate their influence on the bioactive compound’s extraction: accelerated solvent extraction (ASE), ultrasound-assisted extraction (UAE), and laser irradiation extraction (LE). The applied extraction parameters for each technique were chosen based on the results obtained in previous research [19].

#### 3.2.1. ASE

The accelerated solvent extraction of *C. mas* and *C. monogyna fruits* was performed using a Dionex ASE 350 System (Thermo Scientific, Waltham, MA, USA). The extraction parameters were controlled automatically through a control panel. The stainless-steel cells (100 mL) in which a cellulose filter was placed were uploaded with 15 g of each type of fruit that was dried and ground and diatomaceous earth, and the ASE conditions were set as follows—solvent: ethanol solution (50%, *v*/*v*); temperature: 80 °C; static time: 10 min; and number of cycles: 3. The extracts were collected in 250 mL glass vials and stored at 4 °C. According to the ASE extracts’ volume, the extracts’ concentration was 9% (*w*/*v*).

#### 3.2.2. UAE

The UAE process was performed using an ultrasonic bath (Transsonic model T460 H, Elma, Germany) at a working frequency of 35 kHz. Ground plant material was mixed with 50% (*v*/*v*) aqueous ethanol solutions and immersed in the ultrasonic bath for 30 min at 80 °C. The herbal’s mass concentration in the solvent was similar to that of ASE extract: 9% (*w*/*v*).

#### 3.2.3. LE

LE was carried out in similar conditions with ASE: dry and ground fruits were mixed with ethanol solution (50%, *v*/*v*) at an extract concentration of 9% (*w*/*v*) and extracted for 30 min in an extractor with laser radiation at 1550 nm and combined wavelengths of 1270 and 1550 nm, respectively. The LE steel extractor was provided by the Apel Laser S.R.L. (Mogoșoaia, Romania) (Figure 4).

After extraction, the extracts were microfiltrated through a Millipore membrane (0.2 µm pores) and kept in a freezer at −20 °C for further analysis.

### 3.3. Quantification of Polyphenolic Compounds

#### 3.3.1. Spectrophotometric Assay for Total Polyphenols and Flavonoids

Total polyphenols were quantified using the Folin–Ciocalteu method [52]. Briefly, 3 mL of extract and 3 mL of Folin–Ciocalteu reagent were mixed and filtered; 0.5 mL of the filtrate was mixed with 9.5 mL of sodium carbonate, and absorbance was read at 760 nm. The results are expressed as chlorogenic acid equivalent (CAE) calculated using a calibration curve (y = 0.0016x + 0.013; R^2^ = 0.9945).

Total flavonoid content was measured as described earlier [53]. An amount of 2 mL of extract was mixed with 3 mL of methanol and filtrated. An amount of 1 mL of filtrate was mixed with 1 mL of sodium acetate solution (1 mL), aluminum chloride solution (0.6 mL), and methanol (2.4 mL). The absorbance was measured at 430 nm and the flavonoid content was calculated based on a rutin calibration curve (y = 0.0073x − 0.0357; R^2^ = 0.9977).

The yield of extraction was calculated using the following formula:

yield (%) = m_1_/m_2_ × 100, where m_1_ is the mass of dry extract and m_2_ is the mass of the sample.

#### 3.3.2. HPLC-DAD Analysis

HPLC was performed using a Shimadzu system (Shimadzu, Kyoto, Japan) with two LC-20AD pumps, a DGU-20A degassing unit, a SIL-20AC autosampler, a CTO-20A column oven, and an SPDM20A photodiode array detector (PDA). PDA detector absorbance values were recorded in the 200–600 nm range, and peak wavelengths were selectively chosen depending on the maximum absorption of each analyte of interest.

The identification and quantification of vitamin C were performed using the wavelength where vitamin C presents maximum absorption: 243 nm; a Kromasil 100-10-C18 4.6 × 250 mm column; an isocratic elution of mobile phase, 30:70 (5 mmol L-1 sodium 1-heptane-sulfonate in methanol, solvent A; and acetic acid (1%), solvent B); and a flow rate of 1 mL min^−1^.

The analysis of polyphenolic compounds was accomplished on a C18 column, Luna Phenomenex, 100-10, 4.6 × 250 mm, with a mobile phase consisting of water with formic acid and MeCN/methanol (50:50) with formic acid. An elution (0–5 min for 5% solvent B, 5–25 min for 5–30% solvent B, 25–30 min for 30–25% solvent B, 30–38 min for 25% solvent B, 38.01–40 min for 30% solvent B, 40.01–57 min for 40–50% solvent B, 57–58 min for 50% solvent B, 58–60 min for 50–5% solvent B and 60–70 min 5% solvent B) and a flow rate gradient (0–5 min for 1 mL min^−1^, 5–15 min for 1.5 mL min^−1^, 15–35 min for 1 mL min^−1^, 35–40 min for 1–1.5 mL min^−1^, 40–45 min for 1.5 mL min^−1^, 45–47 min for 1 mL min^−1^, 47–50 min for 0.75 mL min^−1^ and 50–70 min for 1.5 mL min^−1^) of the mobile phase were used.

The quantification of anthocyanins and anthocyanidins was achieved through a C18 Kromasil column, 100-10 4.6 × 250 mm, with a mobile phase composed of 5% formic acid in water and 5% formic acid in methanol. An elution (0–15 min for 6–10% solvent B, 15–35 min for 10–50% solvent B, 35–36 min for 50–6% solvent B, 36–45 min for 6% solvent B) and a flow rate gradient (0–29 min for 1 mL min^−1^, 29–33 min for 0.75 mL min^−1^, 33–45 min for 1 mL min^−1^) of the mobile phase was used.

### 3.4. Antioxidant Assays

#### 3.4.1. DPPH Radical Scavenging

The DPPH method was carried out according to Bondet et al. [54] with some modifications. Briefly, in 1.9 mL of methanol, 0.1 mL of extract was added with various concentrations and 1000 µL of DPPH (0.25 mM), and the absorbance was read at 517 nm. The results are presented as inhibition in IC_50_ (μg/mL). Ascorbic acid was used as the standard.

#### 3.4.2. Reducing Power Assay

The reducing power is based on the reduction of Fe^3+^ to Fe^2+^ and was accomplished by a method outlined by Berker [55]. The sample extracts (0.1 mL with varying concentrations) were mixed with 2.5 mL of sodium phosphate buffer (0.2 M) and 2.5 mL of potassium ferricyanide (1%) and incubated at 50 °C for 20 min. Afterward, 2.5 mL of trichloroacetic acid was added to the mixture. An aliquot of 2.5 mL of the mixture was combined with 2.5 mL of water and 0.5 mL of iron chloride solution and the absorbance was read at 700 nm. Ascorbic acid was used as the standard. The increase in absorbance of the reaction mixture indicates an increase in reducing power.

### 3.5. Antidiabetic Assay

#### 3.5.1. α-Amylase and α-Glucosidase Inhibitory Activities

The α-amylase inhibitory effect was evaluated according to our previous study [56]. Sample extracts of 100 μL were mixed with 250 μL of α-amylase from hog pancreas (EC 3.2.1.1) solution in phosphate buffer (pH 6.9) and maintained at 37 °C for 15 min. Then, 250 μL of starch solution (1% *w*/*v*) was introduced and reincubated at 37 °C for 15 min. Subsequently, 500 μL of 3,5-dinitrosalicylic acid reagent was added to stop the reaction, the mixture was heated at 90 °C for 5 min and then cooled at room temperature, and 5 mL of distilled water was added. The absorbance was read at 540 nm.

The α-glucosidase inhibitory activity was estimated as described by Ranilla et al. with some modification [57]. Equal volumes (100 μL) of samples at different concentrations and the yeast α-glucosidase solution (1 U/mL) were mixed with 650 μL of phosphate buffer (0.1 M, pH 6.9) at 37 °C for 15 min. After that, 100 μL of NPG substrate solution (5 mM) was added, and the mixture was incubated at 37 °C for 15 min. Finally, 1000 μL of 0.2 M sodium carbonate solution was added to the mixture to finish the reaction, and the absorbance was recorded at 405 nm. Acarbose was used as the standard drug.

#### 3.5.2. Insulin Secretion Assay

In vitro, the investigation of antidiabetic activity was realized on a murine insulinoma cell line (β-TC-6). β-TC-6 pancreatic beta cells were cultured in normal (5.6 mM) and hyperglycemic (16.7 mM) conditions in the presence of plant extracts, and the effect of stimulating insulin secretion of these extracts was then analyzed. β-TC-6 cells were cultivated in DMEM supplemented with 10% FBS and 1% antibiotic mixture at 37 °C and 5% CO_2_.

β-TC-6 cells were seeded in a 24-well plate at a density of 1 × 10^5^ cells/mL. A period of 24 h after seeding, β-TC-6 cells were stimulated with 5.6 mM (normal condition) and 16.7 mM glucose (hyperglycemic condition) in the presence or absence of the samples of interest (at a concentration of 250 and 500 μg/mL) for 1 h at 37 °C. A Mouse Ins1/Insulin-1 ELISA Kit was used to quantitatively measure insulin secretion [58]. L-alanine (10 mM) was the reference stimulant of insulin secretion from pancreatic beta cells [59].

### 3.6. Statistical Analysis

Three independent experiments were carried out and the obtained data are presented as the mean ± standard deviation (SD) (*n* = 3). The sample pair of interest was analyzed using the paired Student’s *t*-test (Microsoft Excel 2018 software). Significant statistical differences were considered as *p* < 0.05. The obtained data were processed using statistical procedures to highlight any significant relationship between the chemical composition of the extracts and biological activities.

## 4. Conclusions

The present study demonstrated the impact of green extraction methods on the bioactive compounds and biological activities in *C. mas* and *C. monogyna* fruit extracts. The ASE and laser irradiation method were efficient in the case of polyphenols and vitamin C compounds from *C. mas* and *C. monogyna* fruit extracts. All samples of *C. mas* fruit extracts had the highest values for antioxidant activity (DPPH and reducing power methods). However, using ASE and LE at combined wavelengths of 1270 and 1550 nm leads to extracts with the highest antiradical (DPPH) and iron-reducing power as well as the highest enzyme inhibition activity against tested enzymes. The extracts obtained by ASE and LE presented a considerable α-amylase and α-glucosidase inhibition, correlated with their polyphenols (phenolic acids, flavonoids, and anthocyanins) and high vitamin C contents. Furthermore, the results from this study pointed out that the extracts obtained by ASE stimulate insulin secretion in vitro. Both extracts showed a strong stimulatory effect of insulin secretion in the in vitro β-TC-6 pancreatic beta cell stimulation model. This study provides data for the efficiency of green extraction methods and also for the effects of *C. mas* and *C. monogyna* fruit extracts and suggests future in vivo studies for obtaining new drugs for the prevention and/or treatment of metabolic disorders.

## Figures and Tables

**Figure 1 molecules-29-03595-f001:**
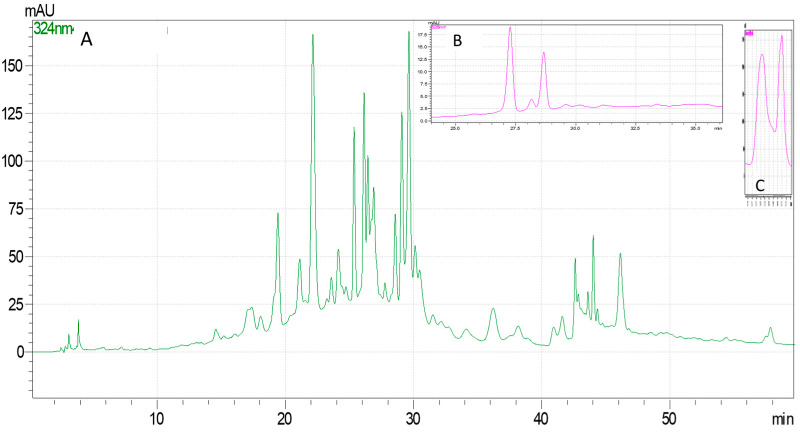
Chromatograms for *C. mas* extract obtained by ASE: (**A**) the chromatogram obtained for the analysis of polyphenolic compounds at 324 nm; (**B**) the chromatogram obtained for the analysis of anthocyanins at 526 nm; (**C**) the chromatogram obtained for the analysis of vitamin C at 248 nm.

**Figure 2 molecules-29-03595-f002:**
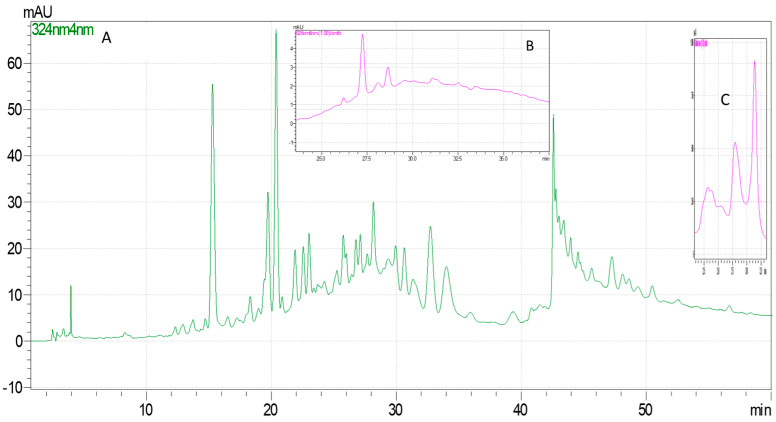
Chromatograms for *C. monogyna* extract obtained by ASE: (**A**) the chromatogram obtained for the analysis of polyphenolic compounds at 324 nm; (**B**) the chromatogram obtained for the analysis of anthocyanins at 526 nm; (**C**) the chromatogram obtained for the analysis of vitamin C at 248 nm).

**Figure 3 molecules-29-03595-f003:**
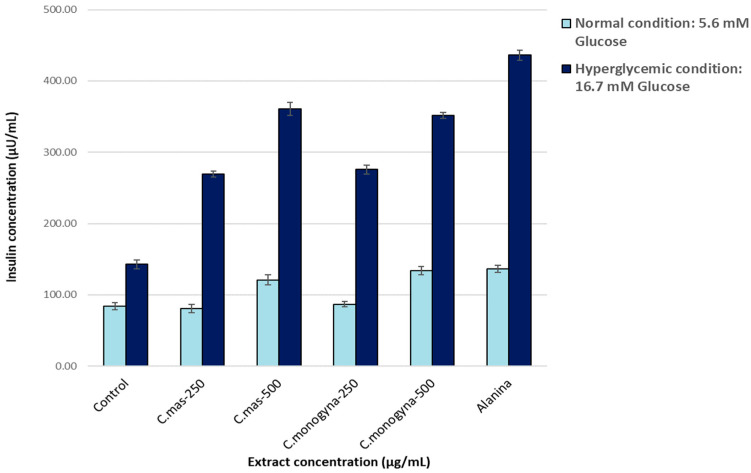
Quantification of insulin secreted by β-TC-6 pancreatic cells following their stimulation and treatment with the samples of interest (ELISA method), in both normal and hyperglycemic conditions. Control—untreated cells. *p* < 0.05, compared to the untreated cells.

**Figure 4 molecules-29-03595-f004:**
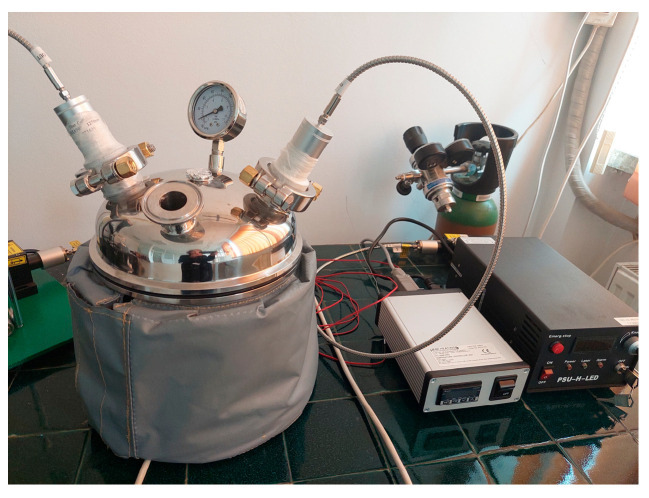
Laser-assisted extraction installation.

**Table 1 molecules-29-03595-t001:** The active biological compound content (polyphenols, flavonoids) and yield of the extracts of *C. mas* and *C. monogyna* prepared using different extraction methods.

Sample	Extraction Method	Polyphenols (mg CAE/g Dry Extract)	Flavonoids (mg RE/g Dry Extract)	Yield (%)
*C. mas*	ASE	103.94 ± 2.60	2.57 ± 0.02	36.1 ± 1.60
	UAE	79.46 ± 0.20	0.95 ± 0.01	21.8 ± 0.90
	LE_1550_	81.38 ± 0.60	0.92 ± 0.01	30.6 ± 1.60
	LE_comb._	85.33 ± 0.40	1.44 ± 0.01	31.8 ± 1.40
*C. monogyna*	ASE	84.89 ± 0.60	3.02 ± 0.03	48.4 ± 2.20
	UAE	54.87 ± 0.40	2.04 ± 0.02	18.2 ± 0.70
	LE_1550_	57.31 ± 0.50	1.98 ± 0.01	19.5 ± 0.80
	LE_comb._	76.59 ± 0.60	2.25 ± 0.02	20.2 ± 1.10

CAE—chlorogenic acid equivalent; RE—rutin equivalent; LE_1550_—laser irradiation at 1550 nm; LE_comb_—laser irradiation at 1550 + 1270 nm. Results are expressed as mean ± SD (*n* = 3).

**Table 2 molecules-29-03595-t002:** Contents of target compounds in the *C. mas* extracts (µg/g dry extract).

Compound	ASE	UAE	LE_1550_	LE_comb._
Gallic acid	76.11 ± 3.80	74.83 ± 2.60	73.37 ± 3.20	74.90 ± 4.30
Chlorogenic acid	504.72 ± 11.20	415.70 ± 9.20	362.09 ± 11.30	430.19 ± 12.70
Caffeic acid	1020.56 ± 14.90	710.07 ± 11.40	701.63 ± 15.80	781.76 ± 17.10
Coumaric acid	39.44 ± 1.80	32.89 ± 1.20	33.76 ± 1.90	38.93 ± 1.50
Rutin	957.22 ± 12.60	528.52 ± 12.30	533.01 ± 11.70	635.85 ± 10.40
Ellagic acid	118.18 ± 4.50	97.80 ± 4.30	105.90 ± 3.70	130.20 ± 5.20
Quercetin 3-β-D-glucoside	80.00 ± 3.10	27.18 ± 0.90	18.82 ± 0.80	20.69 ± 1.20
Quercitrin	183.89 ± 9.50	60.74 ± 1.60	26.47 ± 1.90	71.38 ± 4.80
Myricetin	45.28 ± 2.40	15.10 ± 0.80	18.11 ± 0.90	22.89 ± 1.60
Quercetin	18.33 ± 1.10	14.43 ± 0.50	12.89 ± 0.60	16.67 ± 0.90
Luteolin	20.00 ± 1.70	4.70 ± 0.30	4.63 ± 0.30	5.57 ± 0.30
Kaempferol	7.78 ± 0.60	nd	1.76 ± 0.10	1.82 ± 0.10
Isorahmentin	nd	nd	5.88 ± 0.20	5.90 ± 0.20
Delphinidin-3-glucopyranoside	33.06 ± 1.30	32.21 ± 1.20	33.27 ± 2.40	36.33 ± 1.90
Cyanidin-3-glucopyranoside	566.94 ± 11.50	248.32 ± 3.80	405.88 ± 4.60	433.32 ± 8.20
Petunidin-3-glucoside	237.78 ± 4.90	110.74 ± 2.70	170.59 ± 5.30	182.70 ± 4.40
Delphinidin	12.22 ± 0.90	11.74 ± 0.80	11.76 ± 0.80	11.01 ± 1.10
Cyanidin	11.67 ± 0.70	11.07 ± 0.60	11.11 ± 0.50	10.69 ± 0.90
Vitamin C	22,379.17 ± 16.40	18,819.46 ± 15.30	15,778.43 ± 14.10	18,655.35 ± 16.20

LE_1550_—laser irradiation at 1550 nm; LE_comb_—laser irradiation at 1550 + 1270 nm; nd—not detected.

**Table 3 molecules-29-03595-t003:** Contents of target compounds in the *C. monogyna* extracts (µg/g dry extract).

Compound	ASE	UAE	LE_1550_	LE_comb._
Gallic acid	59.13 ± 3.50	52.20 ± 3.20	50.26 ± 2.90	52.97 ± 4.10
(+)-Catechin	3262.81 ± 14.20	3192.31 ± 13.70	4012.82 ± 14.50	4276.73 ± 15.20
Chlorogenic acid	492.77 ± 8.30	489.56 ± 7.50	507.18 ± 8.10	533.66 ± 8.30
Caffeic acid	28.39 ± 1.40	26.37 ± 1.20	27.69 ± 1.40	30.30 ± 2.60
(−)-Epicatechin	3841.94 ± 15.80	3679.67 ± 16.10	3750.26 ± 16.70	4234.16 ± 15.90
p-Coumaric acid	12.50 ± 0.90	14.84 ± 1.10	11.33 ± 0.80	13.88 ± 0.90
Rutin	2614.88 ± 14.60	2312.09 ± 13.20	2720.00 ± 14.60	2927.23 ± 15.80
Ellagic acid	518.18 ± 10.50	497.80 ± 9.30	495.90 ± 8.70	530.20 ± 11.20
Quercetin 3-β-D-glucoside	40.08 ± 3.10	31.87 ± 2.80	28.21 ± 2.30	28.29 ± 2.50
Quercitrin	489.63 ± 9.20	421.98 ± 8.40	432.31 ± 8.10	504.46 ± 9.70
Myricetin	27.98 ± 1.80	25.27 ± 1.20	20.00 ± 1.10	20.79 ± 0.90
Quercetin	50.01 ± 4.30	46.15 ± 2.90	68.72 ± 4.60	66.83 ± 3.30
Luteolin	13.43 ± 0.70	7.14 ± 0.50	9.74 ± 0.70	5.45 ± 0.20
Kaempferol	15.99 ± 1.10	15.38 ± 1.20	14.87 ± 0.90	15.36 ± 1.40
Isorahmentin	3.51 ± 0.20	nd	nd	4.95 ± 0.30
Delphinidin-3-glucopyranoside	18.60 ± 1.40	12.86 ± 0.70	nd	19.20 ± 1.80
Cyanidin-3-glucopyranoside	232.15 ± 3.90	203.85 ± 3.50	203.08 ± 3.70	231.19 ± 4.10
Delphinidin	17.23 ± 1.10	19.78 ± 1.80	18.46 ± 1.40	17.82 ± 1.20
Cyanidin	19.71 ± 1.60	16.78 ± 0.90	18.46 ± 1.20	17.82 ± 1.30
Vitamin C	2593.80 ± 14.80	1732.97 ± 12.70	1594.36 ± 12.30	1521.78 ± 11.90

ASE—accelerated solvent extraction; UAE—ultrasound-assisted extraction; LE_1550_—laser irradiation at 1550 nm; LE_comb_—laser irradiation at 1550 + 1270 nm; nd—not detected.

**Table 4 molecules-29-03595-t004:** Some performance characteristics of HPLC-DAD methods for phenolic compounds, anthocyanins, and vitamin C analysis.

Compound	λ_max_ (nm)	t_R_ (min)	The Linear Regression Equations	R	Linearity Range of Response (µg mL^−1^)	LoD (µg mL^−1^)	LoQ (µg mL^−1^)
Gallic acid	271	8.2	A = 22,548.51XC − 16,730.64	0.9994	0.5–50	0.11	0.23
(+)-Catechin	279	19.3	A = 8300.645XC − 4379.833	0.9998	0.5–50	0.45	0.47
Chlorogenic acid	327	20.6	A = 62,237.79XC − 17,377.63	0.9998	0.5–50	0.13	0.36
Caffeic acid	324	22.7	A = 117,248.9XC − 27,352.26	0.9999	0.5–50	0.1	0.9
(-)-Epicatechin	279	23.9	A = 10,134.4XC − 1638.931	0.997	0.5–50	0.16	0.30
p-Coumaric acid	310	28.4	A = 174,370.4XC − 39,107.04	0.9993	0.5–50	0.15	0.42
Rutin	354	32.3	A = 31,257.94XC − 4695.0	0.9998	0.5–50	0.24	0.48
Ellagic acid	367	33.8	A =30,052.11XC − 34,356.48	0.9997	0.5–50	0.23	0.34
Quercetin 3-β-D-glucoside	354	35.1	A = 38,616.38XC − 7273.773	0.9996	0.5–50	0.18	0.42
Quercitrin	349	42.7	A = 28,998.06XC − 9131.278	0.9998	0.5–50	0.33	0.45
Myricetin	372	43.3	A= 46,639.07XC − 10,691.48	0.9998	0.5–50	0.11	0.38
Quercetin	372	49.3	A = 111,657.0XC − 44,094.22	0.9998	0.5–50	0.32	0.48
Luteolin	349	50.1	A = 102,976.7XC − 2735.935	0.9996	0.5–50	0.26	0.46
Kaempherol	367	56.7	A = 86,346.8XC − 23,246.12	0.9998	0.5–50	0.15	0.48
Isorahmentin	372	57.8	A = 74,331.04XC − 5962.092	0.9999	0.5–50	0.19	0.47
Delphinidin-3-glucoside	526	25.5	A = 19,024.87XC − 13,742.13	0.9990	0.5–50	0.22	0.41
Cyanidin-3-glucoside	520	27.4	A = 22,562.61XC − 74,68.727	0.9997	0.5–50	0.23	0.38
Petunidin-3-glucoside	526	28.4	A = 32,168.98XC − 7199.806	0.9998	0.5–50	0.19	0.29
Delphinidin	531	31.2	A = 76,198.99XC − 24,569.97	0.9994	0.5–50	0.11	0.30
Cyanidin	526	34.1	A = 55,174.78XC − 17,679.81	0.9993	0.5–50	0.35	0.38
Vitamin C	248	2.7	A= 53,840.48C − 47,183.38	0.9999	1–250	0.2	0.5

**Table 5 molecules-29-03595-t005:** Antioxidant activity of analyzed extracts.

Sample	Extraction Method	DPPH	Fe(III) Reducing Power
		IC_50_ (μg/mL)	
*C. mas*	ASE	31.82 ± 0.1 **	33.95 ± 0.2 **
	UAE	54.70 ± 0.4 **	53.55 ± 0.5 *
	LE_1550_	36.77 ± 0.2 **	52.63 ± 0.3 **
	LE_comb._	39.38 ± 0.3 **	35.33 ± 0.1 **
*C. monogyna*	ASE	522.64 ± 8.4 **	73.04 ± 0.6 **
	UAE	573.91 ± 6.9 *	224.32 ± 2.4 **
	LE_1550_	493.19 ± 7.2 **	369.72 ± 3.1 *
	LE_comb._	469.44 ± 4.8 **	163.26 ± 1.8 **
Ascorbic acid		59.38 ± 0.5	149.48 ± 2.6

ASE—accelerated solvent extraction; UAE—ultrasound-assisted extraction; LE_1550_—laser irradiation at 1550 nm; LE_comb_—laser irradiation at 1550 + 1270 nm. The results are expressed as mean ± SD (*n* = 3); ** *p* < 0.001; * *p* < 0.05 compared with the positive controls.

**Table 6 molecules-29-03595-t006:** α-amylase and α-glucosidase enzyme inhibition of analyzed extracts.

Sample	Extraction Method	α-Amylase Inhibition	α-Glucosidase Nhibition
		IC_50_ (μg/mL)	
*C. mas*	ASE	0.44 ± 0.02 **	77.1 ± 3.10 **
	UAE	96.99 ± 2.70 *	134.5 ± 6.20 *
	LE_1550_	8.51 ± 0.40 *	129.8 ± 5.80 *
	LE_comb._	0.11 ± 0.01 *	98.2 ± 4.70 *
*C. monogyna*	ASE	0.53 ± 0.01 **	108.4 ± 9.40 *
	UAE	7.67 ± 0.90 *	1157.6 ± 10.40 **
	LE_1550_	19.70 ± 1.20 *	181.3 ± 5.30 *
	LE_comb._	1.26 ± 0.01 *	129.1 ± 3.10 *
Acarbose		8.12 ± 0.60	168.4 ± 6.90

Results are expressed as mean ± SD (*n* = 3); * *p* < 0.05 and ** *p* < 0.001; the α-amylase and α-glucosidase inhibition activity compared with polyphenols and flavonoid contents.

## Data Availability

Data are contained within the article.

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
