# Peer review of "Antioxidant and Antidiabetic Activity of Cornus mas L. and Crataegus monogyna Fruit Extracts"

_molecules, 2024, doi:10.3390/molecules29153595_

Round 1

Reviewer 1 Report

Comments and Suggestions for Authors

Paun et al. aimed to evaluate the influence of three extraction techniques on the content of total phenols, total flavonoids, individual phenolic compounds content, antioxidant and antidiabetic activity. In general, the article is well written and discussed in relation to the results obtained and the order followed by the authors. However, from my point of view I would like to highlight several important things about the structure and importance of the techniques studied that should be modified in the article.

1. Introduction section: significance of selected extraction techniques should be more highlighted.

2. The abbreviations should be defined just the first time they are mentioned in the text and used as such throughout the manuscript.

3. Table 1. Statistical analysis should be included, as well as SD values for yield (with the assumption that the measurement was done in triplicates). SD values and statistical analysis should be added in Table 2 and 3.

4. More information about laser irradiation as extraction techniques should be provided and the corresponding results should be more justified.

5. I think it is true that the total phenolic and flavonoid content has been done, but a quantification of only 17 and 20 phenolic compounds has been done by HPLC-DAD. I think it is necessary to do a complete identification of phenolic compounds by using HPLC-MS. This aspect is very important because it is necessary to do a complete characterization, or based on previous studies, you can discuss that a quantification of the most concentrated phenolic compounds has been done.

6. Subsection 3.1. More information about drying process, moisture content and particle size should be provided.

7. Subsection 3.2. On what basis did the author choose the applied extraction parameters for each technique? How much plant material was used for UAE?

8. Subsection 3.3.1. Calibration curve and correlation factor should be provided for total phenolic and total flavonoid contents. These contents were expressed per dry extract or dry sample?

Author Response

Dear reviewer,

Thank you very much for your comments and professional advice. Based on your suggestion and request, we have made corrected modifications to the revised manuscript and I highlighted them in yellow. We have tried our best to improve the manuscript as the reviewer suggested. Furthermore, we would like to show the details as follows:

  1. We have introduced additional information in lines 65-73.
  2. Thank you for pointing this out. I defined the abbreviation in the first part of the manuscript (Lines 62-63 and 106-108) and used it as such throughout the manuscript.
  3. I agree with this comment and I corrected it.
  4. We added more information about laser irradiation as extraction techniques in lines 69-71, 124-126, and 144 – 155.
  5. Although we did not mention it in the article, we initially scanned the extracts by HPLC-MS and later quantified the main compounds in the extract by HPLC-DAD. The quantified compounds were also identified as the main compounds for the extracts studied by other researchers [30, 40].
  6. We added more information about the drying and grounded process in Lines 317-319 (Subsection 3.1.).
  7. Therefore, we have added additional information (Lines 323-324). The quantity of plant material for UAE appears in Line 329 (15 g of dried and ground in a stainless-steel cell).
  8. I agree with this comment and I introduced this information (Lines 356-357). For total flavonoid contents this information already exists (Line 362). These contents were expressed per dry extract.

We would like to thank you for your review, for raising important points, and for providing corresponding suggestions to improve the quality of our paper.

Yours Sincerely,

Dr. Gabriela PAUN

Reviewer 2 Report

Comments and Suggestions for Authors

The authors presented a manuscript titled "Antioxidant and antidiabetic activity of Cornus mas and Crataegus monogyna fruit extracts", which concerns the use of special and automated methods for the extraction of common herbal materials, and it is prepared correctly.

In the manuscript, the authors experimentally tested the impact of green extraction methods on the efficiency of obtaining mixtures of bioactive compounds from C. mas and C. monogyna fruits and demonstrated highly promising biological activity of the obtained extracts, and suggests future in vivo studies for obtaining new drugs for the prevention and/or treatment of metabolic disorders,  and the obtained results suggest the need for future in vivo studies to obtain new drugs for the prevention and/or treatment of metabolic disorders.

After all, the obtained results are consistent with the state of knowledge, including the knowledge of folk medicine, but such good effectiveness of the techniques used is worth publishing in the journal "Molecules" after corrections and removal of several shortcomings, mainly related to the style of the manuscript, listed below:

·       Please make sure to standardize the spacing between lines, unless they are the result of improper text processing in the editorial office.

·       In line 64, please remove the unnecessary dot before the cited literature. It is … compounds. [18, 19]. … , should be … compounds [18, 19]. … .

·       At line 137 is … Table 2 Contents … , but should be Table 2. Contents … . A dot character is needed after the table number. Comment: See line such 102, 184, and 217.

·       At the body of the Table 2 (line 137) and at the Table 3 (line 139) is … USE … , but maybe use … UAE … . Comment: Please correct or define the abbreviation "USE" if necessary and discuss it in the text.

·       At line 139 is … Table 3 Contents … , but should be Table 3. Contents … . Comment: See line such 102, 184, and 217.

·       At line 166, 281,  is … (-)- epicatechin … , but should be better … (−)- epicatechin … . Comment: The mathematical subtraction sign “ “ is desirable.

·       At line 211 is … antioxidants. [43]. … , but should be … antioxidants [43]. … . Comment: Please remove the dot before cited literature. Similar mistake is at line 243

·       At line 266 is … in the β-TC6 cells.; C. monogyna extract … , but please decode/correct the mark “.; “.

·       At line 280 please add more detail to Sigma-Aldrich customer, such as … Sigma-Aldrich (St. Louis, MO, USA) … . Please provide full details of all manufacturers of chemicals used in the manuscript, including the CO2 supplier, including purity parameters, which in this case may have a key impact on the repeatability of the interesting test results obtained by the authors.

·       At lines 283–284 is … 2,2-difenil-1-picrilhidrazil (DPPH), … , but please check the name of the molecule on the Sigma-Aldrich website and provide the full or simplified correct English name of the molecule.

·       At line 284 is … dinitrosalicylic acid (DNS) … , but maybe it should be … 3,5-dinitrosalicylic acid (DNS) … . Comment: Please check and clarify which dinitrosalicylic acid it is. See line 384 (3.5.1.).

·       At line 333 at the equation the mathematic subtraction sign “ – “ is desired.

·       At line 334 is … 4.6x250 … , at line 348 is … 4.6x250 … , and at line 357 is … 4.6x250 … , but should be at line 344 … 4.6×250 … , at line 348 … 4.6×250 … , and at line 357 … 4.6×250 … , or should be at line 344 … 4.6 × 250 … , at line 348 … 4.6 × 250 … , and at line 357 … 4.6 × 250 … , respectively. Comment: One of the mathematical multiplication signs is required, such as "×", unfortunately the authors use the letter "x". See lines 335 and 506.

·       At line 393 is … Acarbose was used as the standard drug. … , however, please check the meaning/correctness of the information provided and translate it if necessary. Maybe should be better … An acarbose drug was used as the standard. … .

·       At line 400 is … 370C … , but should be … 37°C … . Please use the classic degree sign " ° ". See lines such 382 and 391.

·       At line 401 is … 1x10 … , but should be … 1×10 … .

·       Conclusion paragraph: Green extraction methods are well known and used for the extraction of herbal materials, generally for the extraction of plant materials, for example for the extraction of green tea, see such “Conventional and accelerated-solvent extractions of green tea (camellia sinensis) for metabolomics-based chemometrics” at Journal of Pharmaceutical and Biomedical Analysis, 2017, 145, 604–610 (https://doi.org/10.1016/j.jpba.2017.07.027), for this reason, the information on line 417 ... reported first time in this paper ... , seems to mislead the reader of MDPI, especially since the use of these techniques for the first time by the authors in the authors' laboratories does not deserve information in the scientific work, especially in the conclusions paragraph. Comment: Please remove information that misleads the reader.

·       Below line 440, in the references paragraph, it is desirable to complete the DOI numbers of the articles in order to facilitate work in subsequent stages of production in the editorial office of the Molecules journal.

Author Response

Dear Reviewer,

Thank you very much for your comments and professional advice. Based on your suggestion and request, we have made corrected modifications to the revised manuscript and I highlighted them in yellow. We have tried our best to improve the manuscript as the reviewer suggested. Furthermore, we would like to show the details as follows:

  1. I agree. I revised the paper and I checked the spacing between lines.
  2. Thank you for pointing this out. I corrected Line 64.
  3. I made all corrections in Tables 2 and 3 (Lines 162 and 164).
  4. Thank you for drawing my attention. I corrected USE with UAE.
  5. I agree. I modified (−)- epicatechin in Table 3 and Line 305 with the mathematical sign “−“.
  6. I removed all the dots before citing literature. Thank you.
  7. I corrected Lines 309 and 316: β-TC-6
  8. I agree. I added details of all manufacturers of chemicals used in the manuscript.
  9. Thank you for drawing my attention. I corrected the compound's name.
  10. In 3.3.2. section we change the letter “x” with mathematical signs "×".
  11. At line 448 (former 393) acarbose is considered an “antidiabetic drug and is often used as a reference to which other α-glucosidase inhibitors are compared” (from Sigma Aldrich description).
  12. Thank you for drawing my attention. I corrected lines 455 and 456 (former lines 400, and 401).
  13. I deleted the words “reported first time in this paper” in the Conclusion paragraph.
  14. I agree and I completed the DOI numbers of the articles.

We would like to thank you for your review, for raising important points, and for providing corresponding suggestions to improve the quality of our paper.

Yours Sincerely,

Dr. Gabriela PAUN

Reviewer 3 Report

Comments and Suggestions for Authors

In this study, three extraction methods, including accelerated solvent extraction, ultrasound-assisted extraction, and laser irradiation extraction, were used for the polyphenolic compounds and vitamin C extraction of Cornus mas and Crataegus monogyna fruit extracts. The polyphenols and vitamin C of extracts were quantified using HPLC-DAD, the total phenolic content, flavonoid content, antioxidant activity (DPPH and Reducing power), and antidiabetic activity were studied. The antidiabetic activity was examined by the inhibition of α-amylase and α-glucosidase, and in vitro on a beta TC cell line (β-TC-6). Although it is relatively comprehensive study, some experiments can be supplemented and the manuscript should be revised. Followings are some suggestions for further revisions.

1. Please check and revise the abbreviations throughout the manuscript. For example, in the abstract, the “ASE and laser irradiation (LE)” should be defined their first used in Lines 9-10.

2. In the abstract, the phrase “and could be used to prevent and treat diabetes” can be deleted. Please describe in brief (one sentence) why using “Cornus mas and Crataegus monogyna” as research target, and why studied them together.

3. Please check the plants’ Latin names. For example, the abbreviation form “C. mas” can be used after its full name “Cornus mas L.” had been given.

4. In the introduction, please explain why using two different plants as research object. The introduction section can be written in a more logic way.

5. “For a better comparison of the effectiveness of the 3 methods approached, the following parameters were kept constant: the same solvent (50% (v/v) hydroalcoholic solution), extraction time of 30 minutes, extraction temperature of 80°C, and a solid-liquid ratio of 1:10.” This is not reasonable, as they are three different method. It is suggested to optimized the 3 extraction method separately, and make comparisons with their highest extraction efficiency, respectively.

6. The purity of chemical compounds can be indicated in the 3.1. Materials section.

7. For the HPLC analysis, the quantification method should be validated, such as the linear ranges, LOD, LOQ, and recovery. Please provide typical HPLC chromatograms of the extracts.

8. The inhibition curves for “α-amylase and α-glucosidase inhibition” tests should be provided, as well as for the antioxidant activity test.

Author Response

Dear Reviewer,

Thank you very much for your comments and professional advice. Based on your suggestion and request, we have made corrected modifications to the revised manuscript and I highlighted them in yellow. We have tried our best to improve the manuscript as the reviewer suggested. Furthermore, we would like to show the details as follows:

  1. Thank you for pointing this out. I checked and revised the abbreviations throughout the manuscript.
  2. I deleted the phrase “and could be used to prevent and treat diabetes” from the Abstract. I added a short explanation about why studied the two plants together (Lines 72-75).
  3. I agree. I corrected the full name “Cornus mas L.” Thank you.
  4. I explained in Lines 72-75 why we chose the two plants as research objects.
  5. Although there are different extraction methods, by maintaining fixed extraction parameters (time, temperature, solvent, and plant: solvent ratio) we can consider that the efficiency of the methods depends on their mechanism, as the only variable element. Regarding the optimization of the laser-assisted extraction method, the parameters were selected based on the results obtained in previous research [19].
  6. I agree. I introduced this information in Line 301.
  7. Thank you for pointing this out. The authors agree with the reviewer and we want to inform the reviewer that in our lab when a new method is developed, also it is validated by following some method performance parameters recommended in the guidelines of AOAC, including selectivity, linearity, limit of detection, precision, accuracy, and robustness, and before of any sample set, the performance characteristics of the method are determined. Therefore, some modifications were made, and a table (Table 4) with those was added, please see the manuscript where the results are presented. Also, we have added some chromatograms of the extracts, please see in manuscript.
  8. All IC50 values ​​were calculated based on the inhibition curves of the two enzymes, as well as for the antioxidant activity. It did not seem logical to me to duplicate the information by putting both the IC50 values ​​and the inhibition curves in the article, but at the Reviewer's request, I sent them as a supplementary file.

We would like to thank you for your review, for raising important points, and for providing corresponding suggestions to improve the quality of our paper.

Yours Sincerely,

Dr. Gabriela PAUN

Round 2

Reviewer 1 Report

Comments and Suggestions for Authors

 The authors significantly improved the quality of the manuscript.

Reviewer 3 Report

Comments and Suggestions for Authors

Most of the comments have been addressed. Here are some suggestions for further minor revisions.

1, The ” L.” (Line 11, Line 28, Line 72, Line 75) and “fruit” (Line 19, Line 209, Line 269) can be in normal font style. The “DPPH, and Reducing power” (Line 13) can be changed to “DPPH and reducing power”. Please check similar problems throughout the manuscript.

2, “31.82±0.1” and “33.95±0.2” (Line 18) should have the same significant digits. Can they be changed to “31.82±0.10” and “33.95±0.20”, respectively? Please check similar problems throughout the manuscript, especially in Table 2, Table 3, Table 5, and Table 6.

3, The “Cornus mas L.” (Line 75) and “C. mas L.” (Line 78) can be changed to “C. mas”.

4, The peaks of the determined compounds should be marked in the Figure 1 and Figure 2. The Figure 1 and Figure 2 can be combined into one figure, and marked as A and B.

5, Please check the abbreviations. For example, the “accelerated solvent extraction (ASE), ultrasound-assisted extraction (UAE), and laser irradiation extraction (LE).” (Line 353-354), these abbreviations have been defined in the previous text (Introduction section).

6, The “n” of “n=3” can be in Italics font style.